# Recognition Method for Broiler Sound Signals Based on Multi-Domain Sound Features and Classification Model

**DOI:** 10.3390/s22207935

**Published:** 2022-10-18

**Authors:** Weige Tao, Guotao Wang, Zhigang Sun, Shuyan Xiao, Quanyu Wu, Min Zhang

**Affiliations:** 1School of Electrical and Information Engineering, Jiangsu University of Technology, Changzhou 213001, China; 2Electronic Engineering College, Heilongjiang University, Harbin 150080, China; 3Reliability Institute for Electric Apparatus and Electronics, Harbin Institute of Technology, Harbin 150001, China

**Keywords:** multi-domain sound features, classification model, broiler sound signal, majority voting processing, recognition accuracy

## Abstract

In view of the limited number of extracted sound features, the lack of in-depth analysis of applicable sound features, and the lack of in-depth study of the selection basis and optimization process of classification models in the existing broiler sound classification or recognition research, the author proposes a recognition method for broiler sound signals based on multi-domain sound features and classification models. The implementation process is divided into the training stage and the testing stage. In the training stage, the experimental area is built, and multiple segments of broiler sound signals are collected and filtered. Through sub-frame processing and endpoint detection, the combinations of start frames and end frames of multiple sound types in broiler sound signals are obtained. A total of sixty sound features from four aspects of time domain, frequency domain, Mel-Frequency Cepstral Coefficients (MFCC), and sparse representation are extracted from each frame signal to form multiple feature vectors. These feature vectors are labeled manually to build the data set. The min-max standardization method is used to process the data set, and the random forest is used to calculate the importance of sound features. Then, thirty sound features that contribute more to the classification effect of the classification model are retained. On this basis, the classification models based on seven classification algorithms are trained, the best-performing classification model based on k-Nearest Neighbor (kNN) is obtained, and its inherent parameters are optimized. Then, the optimal classification model is obtained. The test results show that the average classification accuracy achieved by the decision-tree-based classifier (abbreviated as DT classifier) on the data set before and after min–max standardization processing is improved by 0.6%, the average classification accuracy achieved by the DT classifier on the data set before and after feature selection is improved by 3.1%, the average classification accuracy achieved by the kNN-based classification model before and after parameter optimization is improved by 1.2%, and the highest classification accuracy is 94.16%. In the testing stage, for a segment of the broiler sound signal collected in the broiler captivity area, the combinations of the start frames and end frames of multiple sound types in the broiler sound signal are obtained through signal filtering, sub-frame processing, endpoint detection, and other steps. Thirty sound features are extracted from each frame signal to form the data set to be predicted. The optimal classification model is used to predict the labels of each piece of data in the data set to be predicted. By performing majority voting processing on the predicted labels of the data combination corresponding to each sound type, the common labels are obtained; that is, the predicted types are obtained. On this basis, the definition of recognition accuracy for broiler sound signals is proposed. The test results show that the classification accuracy achieved by the optimal classification model on the data set to be predicted is 93.57%, and the recognition accuracy achieved on the multiple segments of the broiler sound signals is 99.12%.

## 1. Introduction

Relevant studies have shown that animal sounds contain much emotional information, such as crows when they are hungry [1], screams when they are in pain [2,3], and weeps when they are separated [4]. Therefore, sounds can be used to provide feedback on their own body conditions and external environment changes, and many scholars have carried out relevant research. For example, Marx et al. [5] studied the voices of chicks in the process of grouping, including distress calls, short peeps, warblers, and pleasure notes, and found that the voices would change with the growth environment and social ability. Zeltner et al. [6] studied the fear reaction (especially the calls) of laying hens composed of the three different strains when they are attacked by a predator. Fontana et al. [7] conducted a study on the peak frequency of broiler sound signals and concluded that there is a significant correlation between the peak frequency of broiler sound signals and its own weight. The more they grow, the lower the frequency of the sounds made by the broilers.

During the growth of broilers, they are easily infected with diseases such as bronchitis, laryngotracheitis, avian influenza, pullorum, etc. The upper respiratory mucosa of these diseased broilers is strongly stimulated, which causes a low cough and makes them sluggish [8]. Therefore, the sounds represented by the cough can provide feedback on the mental state of the broiler and indirectly express their physical condition, that is, their health status. During the investigation of Huixing farm of Liushu Town, Linkou County, Mudanjiang City, Heilongjiang Province, the author and team members learned that experienced breeders used the phenomenon of “the sick chicken makes coughs” and then summarized an effective broiler health judgment method. That is, in the process of distributing feed to each broiler captivity area, the breeders listened to whether the broilers coughed to roughly determine whether there were diseased broilers in the current broiler captivity area. It should be noted that twenty-five to thirty broilers are usually cultivated in each broiler captivity area. In addition, bronchitis, laryngotracheitis, avian influenza, pullorum, and other diseases are contagious. Often, after a broiler is diseased, similar diseases will spread quickly in the broiler captivity area. Therefore, the timely detection of diseased broilers can greatly reduce losses [9].

It can be seen from the above-mentioned broiler health judgment method that accurately capturing and recognizing coughs in the broiler sound signals are key to judging the health of the broilers. In fact, by applying sensor technology and wireless communication technology, we can design an audio collection terminal and place it in the broiler captivity area to collect segments of the broiler sound signal and send it to a remote computer. On this basis, if the coughs can be accurately recognized from the broiler sound signal, it can realize automatic broiler health monitoring. Under normal circumstances, the broiler sound signal contains sound types that represent the growth status or physiological characteristics of the broilers, such as crowing, coughing, purring, and so on. Therefore, the accurate recognition and classification of the sound types contained in the broiler sound signal are key to achieving the accurate recognition of coughs.

Many scholars have conducted a series of studies on the recognition methods for broiler sound signals in different states or recognition methods for different sound types in the broiler sound signals. For example, Lee et al. [10] used the Support Vector Machine (SVM) as a classification model to identify the sound types while the laying hens suffered physical stress from changes in temperature and mental stress from fear. Cheng et al. [11] proposed a new-type chicken voice recognition method. This method used sparse representation to complete the noise reduction and feature extraction from the reconstructed voices. It also used SVM to the classify chicken voices under different environments. After extracting the features from the sound signals of Hailan brown laying hens, Yu et al. [12] analyzed the correlations between the calls and behaviors of the laying hens. In the subsequent research [13], several machine learning algorithms were used to classify and recognize the different sound signals of the laying hens, and the J48 decision tree proved to achieve the highest classification accuracy. Cao et al. [14], in the same research group, used the principle that different sound types have different power spectral densities to carry out research on the classification and identification of the calls and fan noise of Hailan brown laying hens. This research helped to realize the detection and extraction of animal sounds in a noisy environment.

It can be seen from the above studies that the current research on broiler sound signal classification is mainly concentrated on the extraction of sound features, including the extraction of the sound features that can distinguish the different sound types in broiler sound signals and the extraction of the sound features that can distinguish the sounds or calls made by broilers from the environmental noises. On this basis, they built the data set and trained the classification models based on machine learning classification algorithms to complete the recognition of different sound types or different sounds. Although these methods have achieved satisfactory results during the early development process, three problems still exist. The first problem is the extraction of sound features. In these studies, they chose to extract one or a few commonly used sound features from the time domain or the frequency domain from the sound signals to build the data sets without considering the sound features that can be extracted from multi-domains and extracting more sound features. The second problem is the analysis of the sound features. They did not analyze whether the selected sound features are applicable to the current classification problem. In general, the selected general sound features often perform poorly in dealing with personalized problems. The third problem is the training of the classification model. They chose to train the classification model based on a certain machine learning classification algorithm according to experience but did not provide the selection basis. Meanwhile, they did not optimize the inherent parameters of the trained classification model to improve its classification effects further.

It is worth mentioning that, in addition to SVM and the decision tree mentioned above, when combined with some optimization algorithms, new classification methods also show their skills in solving classification problems in various fields, such as support matrix machines, multi-class fuzzy support matrix machines, deep stacked support matrix machines, Twin robust matrix machines, and so on. For example, Xu et al. [15] used support matrix machines, double support matrix machines, and near-end support matrix machines to solve the classification problem of two-dimensional image data. Experiments have shown that the three algorithms achieved stable and efficient classification results. Horng [16] applied a multi-class support vector machine classifier to solve the image classification problem of the supraspinatus muscle and achieved better performance than the other implementation methods. By establishing the objective function of a non-parallel hyperplane and integrating fuzzy attributes, Pan et al. [17] proposed a multi-class fuzzy support matrix machine (MFSMM), which was used to solve the classification problem of two kinds of roller-bearing experimental data. Hang et al. proposed a new deep stacked support matrix machine (DSSMM) to improve the performance of existing shallow matrix classifiers in EEG classification. Pan et al. [18] proposed a new non-parallel classifier called the twin robust matrix machine (TRMM) and applied it to roller-bearing fault diagnosis. The experimental results show that the method has excellent fault diagnosis performance, especially in the presence of abnormal samples.

On the basis of summarizing the previous research results, combined with the shortcomings of the existing broiler sound signal classification research, in this paper, the author proposes a recognition method for broiler sound signals based on multi-domain sound features and classification models. Specifically, in view of the limited sound features extracted in the existing broiler sound signal recognition methods, we extracted a total of sixty sound features from four domains, including the time domain, frequency domain, Mel-Frequency Cepstral Coefficients (MFCC), and sparse representation. These sound features can fully describe the differences between the different sound types in the broiler sound signals. In view of the lack of in-depth analysis of sound features in the existing broiler sound signal recognition methods, we used a random forest to calculate the importance of the sound features and retain those sound features that can effectively reflect the differences among the sound types to build the high-quality data set. In order to solve the problem of lacking in-depth comparison and the parameter optimization of the classification models in the existing broiler sound signal recognition methods, we trained the classification models based on different machine learning classification algorithms, respectively, and compared them to obtain the best performer and optimize its inherent parameters to improve its classification effects. In this way, the optimal classification model can accurately and effectively recognize each sound type in a segment of the broiler sound signal and complete the cough recognition at the same time in this process. This method is an important basis for follow-up research on automatic broiler health monitoring.

We summarize the contributions of this paper in the following:(1)Extract multiple sound features from the four domains of time domain, frequency domain, MFCC, and sparse representation to fully describe the differences between the sound types in the broiler sound signals.(2)Use random forest to calculate the importance of each sound feature and retain those sound features that contribute greatly to the classification effects of the classification model to build a high-quality data set.(3)Train classification models based on different classification algorithms, compare them to obtain the best performer and optimize their inherent parameters to obtain the optimal classification model.(4)Combine the prediction results of the classification model and majority voting processing, provide a recognition process for sound types, and newly propose the definition of the recognition accuracy for broiler sound signals.

## 2. Methods

In order to carry out the research on the recognition method for the broiler sound signals based on multi-domain sound features and a classification model, we divided the whole research process into two stages, as shown in Figure 1. The first is the training stage. In this stage, we used the audio collection system to collect multiple segments of the broiler sound signals in the broiler captivity area and performed signal filtering on the original sound signals. Through sub-frame processing and endpoint detection, we processed each segment of the broiler sound signal into multiple frame signals and obtained the combinations of multiple frame signals corresponding to each sound type. We extracted multiple sound features from each frame signal to construct multiple feature vectors and obtained multiple pieces of data. The labels of the data were manually labeled by determining the sound type corresponding to the combination of each frame signal. In this way, a plurality of labeled data can be obtained from the multiple frame signals contained in the multiple segments of the broiler sound signals, and the data set can be constructed. On this basis, the classification models based on different machine learning classification algorithms were trained, and the inherent parameters of the one with the best performance were optimized to attain the optimal classification model. The second is the testing stage. For a segment of the broiler sound signal collected in the broiler captivity area, after signal filtering, we performed sub-frame processing and endpoint detection to obtain multiple frame signals and the combinations of multiple frame signals corresponding to each sound type. By extracting the sound features from each frame signal, a number of unlabeled data and combinations of unlabeled data corresponding to each sound type can be obtained. We applied the optimal classification model to predict the labels of these data. Through majority voting processing, the common labels of a plurality of unlabeled data corresponding to each sound type were determined. At present, the sound types contained in a segment of the broiler sound signal are recognized; that is, the broiler sound signal recognition method proposed in this paper is completed. The above two stages are described in detail in the rest of this section.

### 2.1. Broiler Sound Signal Collection

The broiler sound signals were collected in the No. 3 breeding greenhouse in the Huixing Farm (44°91′ N, 130°02′ E), Liushu Town, Linkou County, Mudanjiang City, Heilongjiang Province, China, from 19 to 20 August 2022. This experiment was approved by the Laboratory Animal Ethics Committee of Heilongjiang University. The approval number is 20220811007, and the approval date was 18 August 2022. The basic situation of the No. 3 breeding shed is: east–west direction, 122.0 m long, 18.8 m wide, 4.6 m roof height, single-story, and the layout of the shed is four rows of walkways. At the westernmost entrance of the shed, there is a reserved space with a length of 3.2 m and a width of 18.8 m, which is used for walking and placing breeding feed. The author used a fence to enclose an area of 1.2 m long and 2 m wide in the southwest corner as the experimental area and covered the area with 50-to-100-mm-thick wood chips to be used as bedding. This area is about two meters away from the nearest broiler captivity area. It should be noted that when the author directly arranged the audio collection system in the broiler captivity area in the breeding greenhouse, because the broilers are afraid of strangers and electronic equipment, the broilers close to the audio collection system would scream or flap their wings. These abnormal and high-frequency sounds would quickly affect other broilers around, causing the collected signals to not truly reflect the growth conditions of the broilers in the captivity area. Under normal growth conditions, the broilers are in a quiet state, and individual broilers occasionally crow, cough, etc., but are not contagious. However, most of the sound signals collected in these abnormal growth conditions are overlapping broiler sound signals, which are not conducive to the early development of this research. Therefore, the author built an experimental area far away from the broiler captivity area in order to collect continuous single broiler sound signals without overlapping. This is friendly and helpful for the early development of this study.

The specific breed of broiler in the No. 3 breeding greenhouse is the white-feather broiler. Generally, its incubation period is twenty-one days (about three weeks), and the growth period is fifty days (about seven weeks). The test objects selected by the author are thirty white-feather broilers of the same batch of eight weeks old, twenty of which are healthy broilers and ten of which are diseased broilers. They were placed in the experimental area individually in turn, and the audio collection system was used to collect the sound signals. It should be noted that the healthy broilers and the diseased broilers were selected together because only the sound signals of diseased broilers contain coughs. The audio collection system used the chassis of a National Instruments Co., Ltd. (Austin, TX, USA), and its specific model is PXI-1050. For the chassis, the model of the internal controller is PXI-8196, and the model of the sound capture card is NI4472B (8-channel synchronous collection, 24 bits resolution, and 102.4 kS/s sampling frequency). The sound sensor selected was the MPA201 model from BSWA TECHNOLOGY CO., LTD (Beijing, China); the response frequency is 20 Hz to 20,000 Hz, and the sensitivity is 50 mV/Pa. The sound recording software chosen was NI Sound and Vibration Assistant 2010, whose sampling frequency is 32 kHz and the sampling accuracy is 16 bits, and the monophonic collection is performed. The audio collection system collected a segment of the sound signal in a time unit of five minutes, and the interval between two adjacent sound signals was thirty seconds, and the collected sound signals were stored in the computer in the “*.tdms” format. Reference [19] contains the specific sound collection and procession information mentioned above. Finally, the author collected fifty segments of sound signals from the healthy broilers and twenty segments of sound signals from the diseased broilers.

In fact, because the experimental area was relatively far from the broiler breeding area, and the wood chips were laid in advance, the broiler sound signals collected by the audio collection system contained less background noise. In addition, the author copied the collected broiler sound signals to the computer and performed signal filtering based on the Wiener filtering method to obtain sound signals with a high signal-to-noise ratio (SNR) that can be used for sound feature extraction. That is, the broiler sound signals mentioned in the following description, unless otherwise specified, are all sound signals after signal filtering. This part of the research is summarized in another academic article [19], which will not be described in detail here. For each five-minute sound signal, the author performed pulse extraction and endpoint detection on them. By setting the detection threshold to 1.5 times the total energy of the leading non-segment frame [20], the effective signal pulse can be reserved. Through the artificial recognition of the broiler sound signals saved after pulse extraction and endpoint detection, the sound types that can be detected include crows, coughs, purrs, and flapping wings. Among them, the crow is the natural short cheeping during the growth of healthy broilers, and the sound is relatively loud and sharp. The cough is the prolonged abnormal croak of the diseased broilers, and the sound is relatively low. The purr is a low, continuous, undulating sound made when a foreign body is stuck in the throat of the broiler. The flapping wings sound is produced by inciting the friction and vibration between the wings and the air when the broiler is moving. The sound amplitude is large, and the continuous time is long. At present, the signal or the data source conditions for carrying out the research on recognition methods for broiler sound signals based on multi-domain sound features and classification models are available.

### 2.2. Sound Feature Extraction

In “Broiler Sound Signal Collection”, it can be understood that the sound signal processed by pulse extraction and endpoint detection contains four sound types, which indicates that the classification model this research ultimately needs to build is a four-classification model. In this part, the author mainly introduces the sound features from the four aspects of the time domain, frequency domain, MFCC, and sparse representation, which can be used for broiler sound signal classification and recognition.

#### 2.2.1. Time Domain Features

The time domain features refer to some features related to the time characteristics in the process of signal changes with time [21]. Therefore, the broiler sound signal can be analyzed in the time domain. Suppose the sound signal is $x\left(n\right)$, the sound signal of the i-th frame after sub-frame processing is $x_{i}\left(n\right)$, L is the frame length, and $f_{n}$ is the total number of frames.
(1)The short-term energy is used to characterize the energy of the sound signal, which is defined as:(1)$$En\left(i\right)=\sum_{n=0}^{L}x_{i}^{2}\left(n\right),1\leq i\leq f_{n}$$(2)The short-term average zero-crossing rate refers to the number of times each frame signal crosses the horizontal axis (zero value), which is defined as:(2)$$Zcr\left(i\right)=\frac{1}{2}\sum_{n=0}^{L}\left|sgn\left[x_{i}\left(n\right)\right]-sgn\left[x_{i}\left(n-1\right)\right]\right|\;,1\leq i\leq f_{n}$$(3)The short-term autocorrelation function is used to describe the degree of cross-correlation between the sound signal and itself at different time points, which is defined as:(3)$$R_{i}\left(k\right)=\sum_{n=0}^{L-k-1}x_{i}\left(n\right)x_{i}\left(n+k\right),\;1\leq i\leq f_{n}$$In the formula, k is the amount of delay.(4)Since the autocorrelation calculation process involves multiplication and requires a lot of time overhead, the difference calculation method is further used to propose the short-term average amplitude difference, which is defined as:(4)$$D_{i}\left(k\right)=\sum_{n=0}^{L-k-1}\left|x_{i}\left(n+k\right)-x_{i}\left(n\right)\right|,\;1\leq i\leq f_{n}$$(5)Different from the sum of the squares of short-term energy, the author also uses the short-term average amplitude of the absolute value to describe the energy of the sound signal, which is defined as:(5)$$M\left(i\right)=\sum_{n}^{L-1}\left|x_{i}\left(n\right)\right|,1\leq i\leq f_{n}$$

According to the above-mentioned time domain features, the time domain waveform diagram of the four sound types can be obtained, as shown in Figure 2. It can be seen from the figure that the purr can be effectively distinguished from the flapping wings in the time domain, but it is difficult to distinguish the cough and the crow in the time domain, which requires the further introduction of frequency domain features.

#### 2.2.2. Frequency Domain Features

Since the broiler sound signal is a one-dimensional time domain signal, it is difficult to visually see the law of frequency changes. Therefore, we considered using Fourier Transform to transform it into the frequency domain for analysis [22]. Suppose the sound signal is $x\left(n\right)$, and the sound signal of the i-th frame after sub-frame processing is $x_{i}\left(n\right)$. After the Fast Fourier Transform (FFT) [23], *N* is the FFT length.

(1)Spectral entropy describes the relationship between the power spectrum and entropy rate, which is defined as:(6)$$H_{i}=-\sum_{k=0}^{N/2}p_{i}\left(k\right)logp_{i}\left(k\right)$$

In the formula, $p_{i}\left(k\right)$ is the probability density corresponding to the k-th frequency component of the i-th frame.

(2)The spectral centroid is the centroid of the frequency components, which is defined as:(7)$$f_{g}=\frac{\int_{f_{1}}^{f_{2}}f\cdot X\left(f\right)df}{\int_{f_{1}}^{f_{2}}X\left(f\right)df}$$

In the formula, $X\left(f\right)$ is the frequency amplitude spectrum of the signal, $f_{1}$ is the upper cut-off frequency of the spectrum, $f_{2}$ is the lower cut-off frequency of the spectrum, and the frequency range of the sound signal can be seen from the spectral centroid.

(3)The root-mean-square frequency is used to obtain the root-mean-square of the spectrum, which is defined as:(8)$$RMSF=\sqrt{\frac{\sum_{i=N_{1}}^{N_{2}}f_{i}^{2}\cdot S\left(i\right)}{\sum_{i=N_{1}}^{N_{2}}S\left(i\right)}}$$(4)The frequency standard deviation is used to obtain the variance of the spectrum, which is defined as:

(9)$$RVF=\sqrt{\frac{\sum_{i=N_{1}}^{N_{2}}f_{i}^{2}\cdot S\left(i\right)}{\sum_{i=N_{1}}^{N_{2}}S\left(i\right)}-(\frac{\sum_{i=N_{1}}^{N_{2}}f_{i}\cdot S\left(i\right)}{\sum_{i=N_{1}}^{N_{2}}S\left(i\right)}}{)}^{2}$$ In the formula, $S\left(i\right)=|X\left(i\right){|}^{2}$, which represents the energy spectrum of the signal.

According to the above frequency domain features, the frequency domain waveform diagram of four sound types can be obtained, as shown in Figure 3. It can be seen from the figure that the frequency of the cough is about 1300 Hz to 1900 Hz, and the frequency of the crow is about 1000 Hz to 1300 Hz. Therefore, the above two sound types, which cannot be clearly distinguished in the time domain, can be distinguished well in the frequency domain. In summary, five-dimensional time domain features and four-dimensional frequency domain features can be extracted from the broiler sound signals.

#### 2.2.3. Mel-Frequency Cepstral Coefficients

According to the research on human ear hearing mechanisms, the human ear has different hearing sensitivities to sound signals of different frequencies. The Mel frequency domain takes into account the non-linear characteristics of the frequency perception of the cochlear basal mode in the human auditory system, which has high resolution in the low-frequency region and low resolution in the high-frequency region. It is a simple way to realize the auditory perception domain. The MFCC(s) we used are the cepstrum parameters extracted from the Mel frequency domain; that is, the MFCC also takes into account the characteristics of human hearing sensitivity. It first maps the linear spectrum to the Mel non-linear spectrum based on auditory perception and then converts it to the cepstrum [24,25]. Formula (10) converts the frequency to the Mel scale, and Formula (11) reconverts the Mel scale to frequency. The block diagram of the MFCC feature extraction is shown in Figure 4.
(10)$$f_{mel}=2595log_{10}\left(1+f/700\right)$$
(11)$$f=700(10^{\frac{f_{mel}}{2595}}-1)$$

In the research, the 1st to the 13th cepstrum coefficients after Discrete Cosine Transform (DTC) were generally used as the standard 13-dimensional MFCC parameters because they reflected the static characteristics of the sound signal. The dynamic characteristics of the sound signal can be obtained by the difference in the static characteristics. The first-order difference in the static characteristics reflects the changing speed of the sound signal, and the second-order difference reflects the changing acceleration of the sound signal. The author combined the standard MFCC parameters with the first-order difference and the second-order difference to obtain a total of 39-dimensional MFCC feature parameters, that is, thirty-nine MFCC features.

#### 2.2.4. Sparse Representation

The purpose of the sparse representation of the signal is to use as few atoms as possible to represent the signal in a given over-complete dictionary so that a more concise representation of the signal can be obtained; thus, we can more easily obtain the information contained in the signal [26,27]. Therefore, sparse representation is essentially an optimization problem, and the greedy algorithm is the commonly used method [28]. It can be seen from the literature that scholars have conducted several pieces of research on the use of sparse representation to extract signal features. For example, Whitaker et al. [29] applied it to the study of extracting sound features from the sound spectrogram. Cheng et al. [30] used it to complete the noise reduction and feature extraction of chicken sound signals. Li et al. [31] used it to extract the fault features with high information and high value. Therefore, the author also used sparse representation to study the broiler sound signal.

The Matching Pursuit (MP) algorithm is the earliest and most representative greedy algorithm. Its main idea is to use as few atoms as possible to linearly represent the input signal from the given over-complete dictionary based on a certain similarity measurement criterion, thus, achieving an approximation of the input signal [32]. The disadvantage of the MP algorithm is that in the optimization process, a certain atom may be repeatedly selected, resulting in the poor convergence of the algorithm. Therefore, we further introduce the Orthogonal Matching Pursuit (OMP) algorithm. Compared with the MP algorithm, the OMP algorithm orthogonalizes the matched atoms and all the previously selected atoms in each step of the decomposition. In terms of the decomposition effect, both two use the same number of atoms to approximate the original signal. However, in terms of the accuracy of the sparse representation signal, the OMP algorithm is better than the MP algorithm, but the decomposition time of the OMP algorithm is slightly longer than the MP algorithm. We once again introduce the Genetic Algorithm (GA) to optimize the OMP algorithm and finally form the GA–OMP algorithm used in this research. Figure 5 is the flowchart of the GA–OMP algorithm.

When GA–OMP performs signal reconstruction, in order to better describe the time-varying characteristics of the signal, a complete time frequency atomic dictionary is usually used. Since the Gabor dictionary has better time-frequency characteristics [33], we chose it as the over-complete dictionary in this research. The formula is:(12)$$g_{r}\left(t\right)=\frac{1}{\sqrt{s}}g(\frac{t-\mu }{s})cos\left(vt+\omega \right)$$ In the formula, $g_{r}\left(t\right)=e^{-\pi t^{2}}$ is the Gaussian window function, $\gamma =\left(s,\mu ,v,\omega \right)$ is the time frequency parameter, and s is the expansion factor, μ is the translation factor, v is the frequency factor, and ω is the phase factor.

The space of the time-frequency parameters can be separated into $\gamma =\left(\alpha^{j},p\alpha^{j}\Delta \mu ,k\alpha^{-j}\Delta v,i\Delta \omega \right)$. In the formula, $\alpha =2,\Delta \mu =1/2,\;\Delta v=\pi ,\;\Delta \omega =\pi /6,\;0<j\leq log_{2}^{N},\;0\leq p<N2^{-j+1},\;0\leq k<2^{j+1},\;0\leq i\leq 12$, *N* is the number of samples in each frame. Since the Gabor dictionary has good time-frequency characteristics, we can perform feature extraction on its time-frequency parameters and thus obtain four-dimensional time-frequency features, which are called four sparse representation features.

Since sparse reconstruction is required when extracting features, if the number of matching atoms is not determined, although the signal quality after reconstruction is improved, the processing time will be relatively long for computers with average performance. Many scholars limit the number of atoms to between twenty and forty. The author attempted to reconstruct the broiler sound signal with thirty atoms first and found that using thirty atoms can reconstruct the purr very well, but the reconstruction effect on the crow, cough, and flapping wings is not good. Furthermore, fifty and one hundred atoms were selected to reconstruct the signal, then the comparison chart of the reconstructed signals by using different numbers of atoms was obtained, as shown in Figure 6. It can be found that using thirty atoms to reconstruct the purr is better, using fifty atoms to reconstruct the cough is better, and using one hundred atoms to reconstruct the crow and flapping wings is better. Considering that the number of matching atoms of the broiler sound signals must be the same, the author uses thirty, fifty, and one hundred atoms to reconstruct the different sound types, respectively, and further extracted features. Therefore, a total of twelve sparse representation features can be obtained.

### 2.3. Feature Optimization

As mentioned above, the author extracted sixty sound features from the four aspects of the time domain, frequency domain, MFCC, and sparse representation, as shown in Table 1.

As described in Section 2.1, the author collected seventy segments of broiler sound signals in the No. 3 breeding greenhouse using the existing audio collection system. Similarly, we performed signal filtering on these signals based on the Wiener filtering method to obtain the sound signals with high SNR, which can be used for feature extraction. We processed each broiler sound signal into multiple frame signals and used endpoint detection to obtain the start frames and end frames corresponding to the multiple sound types in each broiler sound signal. Then, all of the frame signals between each group of the start frames and the end frames are considered to belong to the same sound type. Then, all of the frame signals between every start frame and end frame can be regarded as belonging to the same sound type. For example, after performing sub-frame processing on the first five-minute sound signal, the author used endpoint detection to obtain the tenth frame as the start frame and the fifty-fifth frame as the end frame of the crow. Then, the frame signals between the tenth and the fifty-fifth were considered to belong to the crow, and 46 pieces of data labeled as “crow” were extracted. In the same way, the author detected the combinations of the start frames and end frames corresponding to multiple sound types from the above-mentioned seventy segments of broiler sound signals, representing the crow, the cough, the purr, and the flapping wing, respectively. Through selection, the author attained multiple combinations of the start frames and end frames and obtained the corresponding labels. Among them, there are 100 crows, 100 coughs, 50 purrs, and 20 flapping wings. This corresponds to 5918 crow frame signals, 4486 cough frame signals, 8955 purr frame signals, and 9532 flapping wing frame signals. Each frame signal can calculate the specific values of sixty sound features to form a piece of data. Finally, the above-mentioned multiple frame signals formed multiple pieces of data to establish the preliminary data set. The specific description of the preliminary data set is shown in Table 2. It should be noted that we divided the data set into a training set (only including training data) and a testing set (only including testing data) according to a 3:1 ratio. The training set was used to train the classification model and optimize its inherent parameters, and the testing set was used to evaluate the classification effects of the classification models.

In order to eliminate the influence of the unit and scale differences between the features, we chose the min–max standardization method to process the preliminary data set [34]. The calculation formula is:(13)$$m=\left(x-x_{min}\right)/\left(x_{max}-x_{min}\right)$$

In the formula, m is the value after standardization, x is the value before standardization, $x_{min}$ is the minimum value of in column, and $x_{max}$ is the maximum value in the column.

Based on the preliminary data set, the author performed min–max standardization processing to obtain the corresponding processed data set and chose the Decision-Tree-based classifier (abbr. DT classifier) under the default parameter configuration to make predictions on the data set before and after the standardization processing. The DT classifier performed ten predictions on each data set, and we took the average of the classification accuracies obtained from the ten predictions as the final prediction result to reduce the random influence. The prediction results are shown in Table 3. It can be seen from the table that the average classification accuracy achieved by the DT classifier on the data set after min–max standardization processing is slightly higher than that on the data set before processing, with an increase of about 0.6%.

Sixty sound features will inevitably bring about the problem of large computational complexity, and there will be some sound features that do not contribute much to the improvement of the classification effects of the classifier (classification model). It may even neutralize the contributions made by those outstanding sound features. Therefore, the author used the model-based feature selection method to filter the sixty sound features. Because a random forest has a mechanism to evaluate the feature importance by using its own classification accuracy [35], the author used it to filter the above-mentioned sixty sound features. The method of using a random forest to calculate the feature importance is as follows:(1)Using the out-of-bag data to calculate the out-of-bag error for each decision tree in the random forest and record it as *err*1.(2)Performing noise interference on a certain feature of the sample in the out-of-bag data, and then calculating the out-of-bag error and recording it as *err*2.(3)Assuming that there are *N* decision trees in the random forest, the importance of this feature can be expressed as:
(14)$$importance=\sum \frac{err2-err1}{N}$$

We take this calculation result as the judgment of the feature’s importance because if the out-of-bag error of a feature before and after the noise interference is large, it means that this feature has a greater impact on the classification accuracy of the sample, which also shows from the side that its importance is higher.

Based on the obtained feature importance, the steps of feature selection are as follows:

**Step 1:** Calculate the importance of each feature and sort in descending order.

**Step 2:** Set one feature to be removed each time, according to the ranking of feature importance, and remove the feature with the worst importance so that a new feature data set can be obtained.

**Step 3:** Use the new feature data set to train a new random forest and recalculate the importance of each feature and rank it.

**Step 4:** Repeat the above process until the optimal feature data set is left.

By setting the lowest threshold of feature importance, the process of removing the sound features with the least importance can be ended. Finally, a total of thirty sound features were retained, and the specific description is shown in Table 4.

According to the feature selection results, which are shown in Table 4, the author retained the feature data in the data set to form the final data set for training the classification model. We also applied the DT classifier under the default parameter configuration to make predictions on the data set composed of thirty sound features and sixty sound features, respectively, and we performed ten predictions to reduce the impact of random errors. The prediction results are shown in Table 5.

It can be seen from the table that when we used the DT classifier to make predictions, whether the obtained single classification accuracy or the obtained average classification accuracy, the classification accuracy achieved by the DT classifier on the data set composed of thirty sound features is higher than that composed of sixty sound features. In general, the classification accuracy achieved by the DT classifier on the data set composed of thirty sound features is about 3.1% higher than that composed of sixty sound features. Therefore, the author retained the selected thirty sound features to construct the final data set for training the classification model.

### 2.4. Training and Optimization of Classification Models

The classification algorithms commonly used in machine learning include SVM [36], decision tree [37], random forest [38], naive Bayes [39], and kNN [40]. With the help of performance evaluation indexes, the author applied the above classification algorithms to train the classification models on the training set and used the trained classification models to make predictions on the testing set. In order to obtain relatively accurate prediction results, each classification model was subjected to ten tests to reduce the impact of random errors. It is worth noting that in the research process, the author also drew multiple time-frequency maps representing the four sound types to build the picture data set and, on this basis, trained the general neural network models, including the Back-Propagation neural network (BP-NN) [41] and the Convolutional neural network (CNN) [42]. Unfortunately, they did not achieve satisfactory prediction results and spent more time obtaining these prediction results.

Table 6 lists the classification accuracies achieved by the above seven classification algorithms, and all of the tests were conducted on the same computer. From the average classification accuracy, we can see that the average classification accuracy obtained by kNN, random forest, SVM, decision tree, naive Bayes, CNN, and BP-NN decreased successively, which were 92.94%, 90.57%, 88.30%, 81.29%, 73.74%, 67.23%, and 64.56%, respectively. Among them, the average classification accuracy obtained by kNN is the highest. The average classification accuracy of the random forest is also higher, but there is a certain gap compared with kNN. The classical SVM is mainly used for binary classification, and thus the obtained average classification accuracy is slightly lower when it is applied to the multi-classification problem. The average classification accuracy obtained by naive Bayes is the worst because naive Bayes is a classification method based on the independent assumption of the feature conditions. However, in the classification and recognition of the broiler sound signals in this research, all of the sound features are not completely independent, and thus the obtained classification accuracy is poor. The average classification accuracy achieved by the two neural network models is significantly lower than that of the other five traditional machine learning classification algorithms, which shows that the neural network model is not suitable for broiler sound signal recognition.

Table 7 shows the precision, recall, and *F*1 value obtained by the five classification algorithms on the four labels corresponding to the four sound types. It can be seen that the precision, recall, and *F*1 value achieved by kNN are relatively higher, all reaching above 88%. After combining the evaluation results of the multiple indexes in Table 6 and Table 7, we finally decided to choose kNN to train the optimal classification model. Next, we will further optimize the inherent parameters of kNN in order to obtain the optimal classification model for broiler sound signal recognition.

We mainly conducted a parameter optimization of the ***n_neighbor***, ***weights***, and ***metric*** of kNN [43]. ***n_neighbor*** refers to the selection of the k value, that is, the selected k number of the labeled data that is closest to the data to be predicted, and the selected labeled data are used for majority voting. The default k value is 5, and the commonly used k value range is from 1 to 15. ***weights*** refers to the weight of the majority voting, and the default setting is “uniform”, which means uniform weight. There is also an optional “distance” here, which means that the weight is equal to the reciprocal of the distance; that is, the labeled data that is closer to the data to be predicted is given a larger majority voting weight. ***metric*** refers to the distance measurement. The selectable values include “*Euclidean*”, “*Manhattan*”, and “*Chebyshev*”, which represent the Euclidean distance, Manhattan distance and Chebyshev distance, respectively. The more commonly used parameter optimization method is Grid Search. After specifying the parameter range, the Grid Search will traverse all of the parameter combinations for optimization until the optimal combination is obtained. Finally, the author obtained the optimal parameter combination of kNN, as shown in Table 8. On this basis, we trained the optimal classification model based on parameter-optimized kNN. Similarly, we introduced the optimal classification model into the testing set for ten predictions and took the average classification accuracy as its prediction result. Table 9 shows the average classification accuracies achieved by the classification model based on kNN before and after the parameter optimization of the testing set. It can be seen that the average classification accuracy achieved by the classification model (optimal classification model) based on the parameter-optimized kNN is 94.16%, which is about 1.2% higher than the former (ordinary classification model).

### 2.5. Majority Voting Processing

Through the above steps, the author finally achieved the optimal classification model based on the parameter-optimized kNN. In this way, when we use the audio collection system to collect a segment of the broiler sound signal randomly, it is unknown how many crows, coughs, purrs, and flapping wings are involved in this broiler sound signal. We also performed signal filtering based on the Wiener filtering method on this broiler sound signal and performed endpoint detection on the processed broiler sound signal to obtain multiple start frames and end frames of the multiple sound types. We regarded the corresponding start frame and end frame as a combination, and then the multiple frame signals are each a combination belonging to the same sound type. That is, the labels of the multiple data formed from one combination all belong to the same sound type. For example, the start frame of the first combination is the tenth frame, the end frame is the fifty-fifth frame, the start frame of the second combination is the sixty-seventh frame, and the end frame is the one-hundred-thirty-fifth frame. That is, the first combination has a total of forty-six frames, which corresponds to the data set that the first forty-six pieces of data belong to the same sound type and have the same label. The second combination has a total of sixty-eight frames, which corresponds to the data set that the forty-seventh to the one-hundred-fifteenth pieces of data belong to the same sound type and have the same label. We used the optimal classification model to predict the labels of the first forty-six pieces of data and determined the common labels of these forty-six pieces of data through majority voting processing, determining the corresponding sound type. Similarly, the corresponding sound type of the sixty-eight pieces of data of the second combination can be determined. In this way, we can first find how many specific sound types are included in the broiler sound signal through endpoint detection and then use the optimal classification model to predict the labels of multiple pieces of data corresponding to each sound type. The common labels were obtained by performing majority voting processing on the predicted labels, and the specific sound types can be recognized. To sum up, by combining the classification model and majority voting processing, we can finally recognize the specific sound types in a segment of the broiler sound signal.

### 2.6. Performance Evaluation Index

In order to evaluate the classification effects of the trained classification model, some evaluation indexes are needed to measure it. The author mainly selected the classification accuracy [44], precision [45], recall [46], and *F* value [47] to evaluate the classification effects achieved by the classification model on the data set.

Suppose the data set is $D=\left\{\left(x_{1},y_{1}\right),\left(x_{2},y_{2}\right),\cdots ,\left(x_{m},y_{m}\right)\right\}$, where $y_{i}$ is the true label corresponding to the data $x_{i}$ and $f\left(x_{i}\right)$ is the label predicted by the classification model f. The classification accuracy can be expressed as the ratio of the number of data that are correctly predicted to the total number of data, which is as follows:(15)$$acc\left(f:D\right)=\frac{1}{m}\sum_{i=1}^{m}I\left(f\left(x_{i}\right)=y_{i}\right)$$ In the formula, I is the indicator function. Additionally, when $f\left(x_{i}\right)=y_{i}$, $I\left(f\left(x_{i}\right)=y_{i}\right)=1$.

For each label (category) in the data set, the confusion matrix [48] of the prediction results is shown in Table 10.

If we want to evaluate the classification effects of each label, we need to calculate the precision and recall. The precision represents the proportion of the true positive data in all of the data that were predicted to be positive, which is defined as follows:(16)$$P=\frac{TP}{TP+FP}\;$$

While recall represents the proportion of the data that are predicted to be positive in all of the true positive data, which is defined as follows:(17)$$R=\frac{TP}{TP+FN}$$

The *F* value is a comprehensive index, which is the harmonic average of the precision rate and recall rate, and it is defined as follows:(18)$$F=\frac{\left(1+\beta^{2}\right)\times P\times R}{\beta^{2}\times P\times R}$$ In the formula, β is used to balance the importance of precision and recall, and when β=1, it is called the *F*1 value.

Similarly, in order to evaluate the prediction effects of the trained classification model for broiler sound signal recognition, an evaluation index is needed to measure it. Referring to the definition of classification accuracy in machine learning, in this study, the author newly proposed recognition accuracy as the evaluation index to evaluate whether the classification model can accurately predict multiple sound types contained in each broiler sound signal.

It is supposed that a segment of the broiler sound signal P contains n sound types, and the true types of each sound type are expressed as $T_{i}\left(i=1,\;2,\;3,\ldots ,n\right)$. For this segment of the broiler sound signal, the data set to be predicted can be established by sub-frame processing, endpoint detection, sound feature extraction, and other steps. The data set contains n sets of data corresponding to n sound types. The predicted label of each data in the data set to be predicted is obtained by applying the classification model. Combined with majority voting processing, the common labels of the n sets of the data are obtained, and the corresponding predicted types of the n sound types are $P_{i}\left(i=1,\;2,\;3,\ldots ,n\right)$. Then, the recognition accuracy can be expressed as the ratio of the number of sound types that are correctly predicted to the total number of sound types, which is defined as:(19)$$S=\frac{1}{n}\sum_{i=1}^{n}I\left(P_{i}=T_{i}\right)$$ In the formula, I is the indicator function. Additionally, when $P_{i}=T_{i}$, $I\left(P_{i}=T_{i}\right)=1$.

## 3. Verification and Analysis

In this part, the author verified the recognition method for the broiler sound signals based on multi-domain sound features and a classification model, including two parts. The first part verifies the classification effects of the classification model. We used the audio collection system to collect forty segments of sound signals from healthy broilers and twenty segments of sound signals from diseased broilers in the No. 3 breeding greenhouse and performed signal filtering on them. Each sound type and their corresponding combinations of multiple frame signals in each broiler sound signal were obtained through sub-frame processing and endpoint detection. By extracting thirty sound features from each frame signal to obtain multiple pieces of data and labeling the data according to the corresponding sound type we obtained the verification data set. Through min–max standardization processing, the validation data set for classification model prediction is finally obtained. After statistics, the validation data set contains 11,982 pieces of data, including 2818 pieces of data labeled “crow”, 2690 pieces of data labeled “cough”, 3205 pieces of data labeled “purr”, and 3269 pieces of data labeled “flapping wing”. We applied the optimal classification model to make ten times predictions on the validation data set, and the obtained average classification accuracy is 93.57%. Compared with the average classification accuracy of 94.16% achieved by the optimal classification model in Table 9, which is only 0.59% lower. This fully proves the generalization and stability of the trained optimal classification model. At the same time, it also proves the feasibility and practicability of the broiler sound signal recognition method proposed in this paper.

The second part verifies the prediction effects of the classification model for sound type recognition in the broiler sound signals. Similarly, we used the audio collection system to collect a segment of the broiler sound signal in the No. 3 breeding greenhouse and performed signal filtering on it. Each sound type and their corresponding combinations of multiple frame signals in this broiler sound signal were obtained through sub-frame processing and endpoint detection. By extracting thirty sound features from each frame signal, multiple pieces of data were obtained, and the validation data set to be predicted was constructed, which was subject to min–max standardization processing. Figure 7 shows the validation data set to be predicted.

We applied the optimal classification model to predict the labels of each data in the validation data set to be predicted and performed majority voting processing on the predicted labels of multiple data corresponding to each sound type to obtain common labels. At present, the predicted types of multiple sound types in the broiler sound signal were obtained. As shown in Table 11, there are eight sound types in the validation data set to be predicted. The first column in the table is the serial number of the sound type, the second column is the number of data corresponding to the sound type, the third to sixth columns provide the prediction results of the classification model for multiple pieces of data, and the seventh column provides the common label of multiple pieces of data obtained through majority voting processing, that is, the predicted types of the corresponding sound type. The author artificially recognized this segment of the broiler sound signal and provided the true types of eight sound types in the eighth column. It can be found that the recognition accuracy achieved by the classification model is 100% for this segment of the broiler sound signal.

In addition, the author used the audio collection system to collect another ten segments of broiler sound signals in the No. 3 breeding greenhouse and performed signal filtering on them. Each sound type and their corresponding combinations of multiple frame signals in ten segments of broiler sound signals were obtained through sub-frame processing and endpoint detection. By extracting thirty sound features from each frame signal to obtain multiple pieces of data, ten verification data sets to be predicted were built, and min–max standardization processing was performed on them. We applied the optimal classification model to predict the labels of each data in the ten validation data sets to be predicted and performed majority voting processing on the predicted labels of multiple pieces of data corresponding to each sound type to obtain multiple common labels. At present, the predicted types of multiple sound types in ten segments of broiler sound signals have been obtained. Table 12 shows the recognition results of the ten segments of the broiler sound signals.

The recognition results show that the recognition is correct, except for a purr in the fourth broiler sound signal that was wrongly recognized as a cough. After calculation, we attained the average recognition accuracy of the optimal classification model based on the parameter-optimized kNN, which was 99.12%. Similarly, in this process, we manually recognized the true type of each sound type in the ten segments of the broiler sound signals as a reference. At present, the author has completed the whole verification process of the recognition method for the broiler sound signals based on multi-domain sound features and the classification model proposed in this paper. The classification model achieved an average classification accuracy of 93.57% and an average recognition accuracy of 99.12%, which fully shows that the classification model has good classification and recognition effects and strongly proves the feasibility and stability of the broiler sound signal recognition method proposed in this paper further.

## 4. Discussion

It is worth noting that Qadri et al. [49,50] advocated the clinical decision support system using a learning algorithm to further improve or reflect the application value of the proposed method. Therefore, shortly after the completion of this study, the author began to design the audio collection terminal placed in the broiler captivity area and visualization software. At present, the audio collection terminal and visualization software have been designed and preliminarily debugged. On this basis, the combination of the audio collection terminal, the broiler sound signal recognition method, and the visualization software can build a clinical decision support system for automatic broiler health monitoring mentioned in the “Introduction”. That is, the audio collection terminal obtains the broiler sound signals in the broiler captivity area and transmits it to the remote computer in a timely manner through wireless communication. The broiler sound signal recognition method completes the task of recognizing different sound types in the broiler sound signals and saves the recognition results locally. The visualization software reads the recognition results for the display. This part also includes the author’s future research content.

It should be noted that the ten predictions performed by the classification model in “Verification and Analysis” showed that a purr was misjudged as a cough in the fourth segment of the sound signal because the two sounded similar. However, the difference between the two is that the former is a persistent and continuous process, which means that a purr will eventually have more frame signals, while the latter is a relatively short process, which means there will be fewer frame signals. In the following research, the author used the audio collection system to obtain multiple segments of the broiler sound signals again and also constructed multiple data sets to be predicted through signal filtering, sub-frame processing, endpoint detection, and feature extraction. The author used the optimal classification model based on parameter-optimized kNN to make predictions on these data sets and combined the majority voting processing to obtain multiple recognition results. The recognition results also show that the classification model has reliable recognition effects, and there is no case that the cough is mistaken for the other three sound types. This shows that the classification model has a high sensitivity to the recognition of coughs, which is conducive to conducting follow-up research on automatic broiler health monitoring. That is, using the currently trained classification model will not miss or incorrectly recognize the coughs in the broiler sound signal, which is extremely important for the stable judgment of broiler health.

It is also worth noting that during the research process, the author found that there is still a small part of noise in the sound signal after signal filtering, but its influence is relatively small, and it can be clearly distinguished from the crow, cough, purr, and flapping wings. Therefore, in future research, on the one hand, the author considers deep signal filtering of this kind of noise, including the research of adaptive filtering algorithm, in order to completely filter out it. On the other hand, the author considers this kind of noise as one specific sound type, then extracts thirty sound features from it to construct the data as well and sets the labels of these data as “noise”. Therefore, the four-classification problem in this research is extended to a five-classification problem for further research. In addition, apart from the five machine learning algorithms listed in Table 7, in the following research, the author considers using an ensemble learning algorithm, such as XGBoost, to replace the existing kNN in order to obtain a classification model with higher prediction accuracy.

## 5. Conclusions

Sound signals can provide feedback on the emotional information of broilers and then feeds back their own physical conditions (health status). Therefore, broiler sound signals can be used to carry out research on automatic broiler health monitoring, and the accurate recognition of the sound types in the broiler sound signals is key to carrying out this research. Existing broiler sound recognition methods fail to extract enough sound features to feed back the characteristics of the broiler sound signal, fail to analyze and select those sound features that effectively represent the differences between different sounds or different sound types, and fail to deeply study the selection basis and optimization process of the classification models. To solve the above problems, the author proposes a recognition method for broiler sound signals based on multi-domain sound features and a classification model. Compared with the implementation process of the classifier in machine learning, the implementation process of the proposed method includes a training stage and a testing stage.

In the training stage, first, we used the audio collection system to collect multiple segments of broiler sound signals and performed signal filtering on these signals based on the Wiener filtering method. Second, we performed sub-frame processing and endpoint detection on the broiler sound signals. In this way, the broiler sound signal was converted into a plurality of frame signals, and the combinations of the start frames and the end frames of each sound type were obtained. Third, we extracted sixty sound features from each frame signal from four aspects, including the time domain, frequency domain, MFCC, and sparse representation, and constructed multiple pieces of data. The data set was constructed by manually labeling multiple pieces of data. Then, we performed min–max standardization processing on the data set and used random forests to calculate the importance of sixty sound features and then retained thirty more important sound features to build the high-quality data set. Finally, we trained the classification models based on seven classification algorithms and obtained the best-performing kNN to optimize its inherent parameters.

In the testing stage, we used the audio collection system to randomly collect a segment of the broiler sound signal in the broiler captivity area and also performed signal filtering, sub-frame processing, and endpoint detection on it. In this way, we can also attain the combinations of the start frames and the end frames of multiple sound types in the broiler sound signal. We extracted thirty selected sound features from each frame signal to build the data set to be predicted and used the optimal classification model to predict the labels of each data. By performing majority voting processing on the prediction results, we achieved the predicted types of each sound type in the broiler sound signal. On this basis, we newly propose the definition of recognition accuracy for broiler sound signals. Multiple verification results show that the classification accuracy achieved by the optimal classification model is 93.57%, and the obtained recognition accuracy is 99.12%. This study is an important basis for the follow-up research on automatic broiler health monitoring and has important reference value for the recognition of animal sound signals in similar fields.

## Figures and Tables

**Figure 1 sensors-22-07935-f001:**
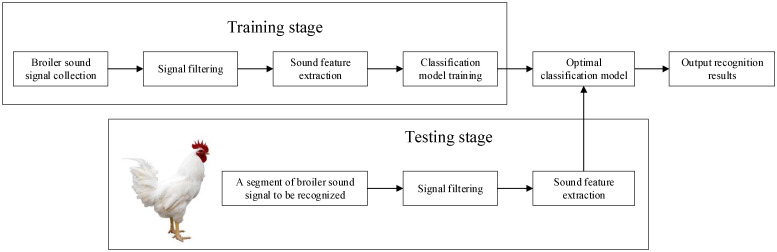
Research process of broiler sound signal recognition method.

**Figure 2 sensors-22-07935-f002:**
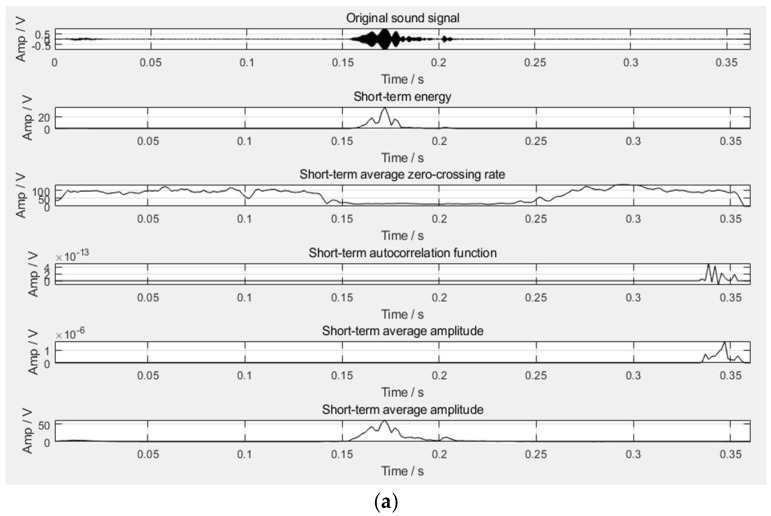
Time domain waveform diagram of four sound types. (**a**) Crow; (**b**) Cough; (**c**) Purr; (**d**) Flapping wings.

**Figure 3 sensors-22-07935-f003:**
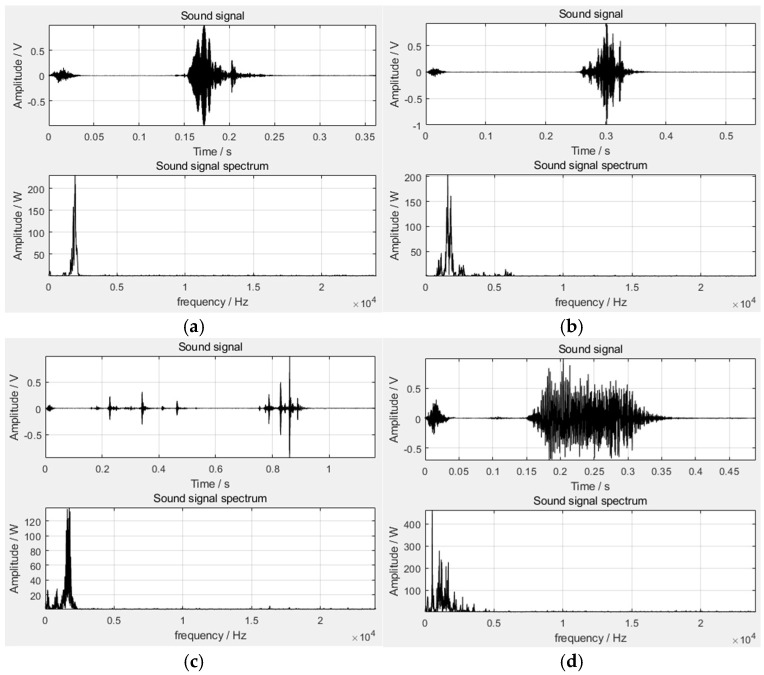
Frequency domain waveform diagram of four sound types. (**a**) Crow; (**b**) Cough; (**c**) Purr; (**d**) Flapping wings.

**Figure 4 sensors-22-07935-f004:**
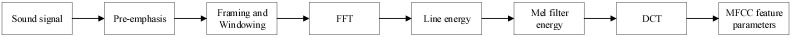
Block diagram of MFCC feature extraction.

**Figure 5 sensors-22-07935-f005:**
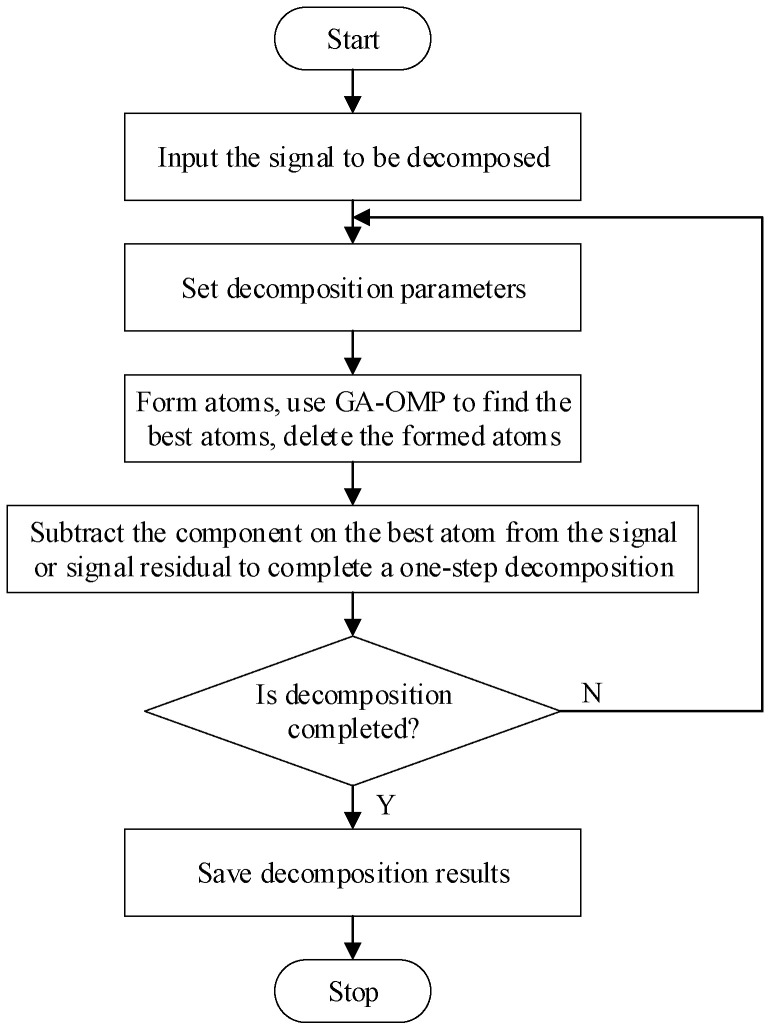
Flow chart of the GA–OMP algorithm.

**Figure 6 sensors-22-07935-f006:**
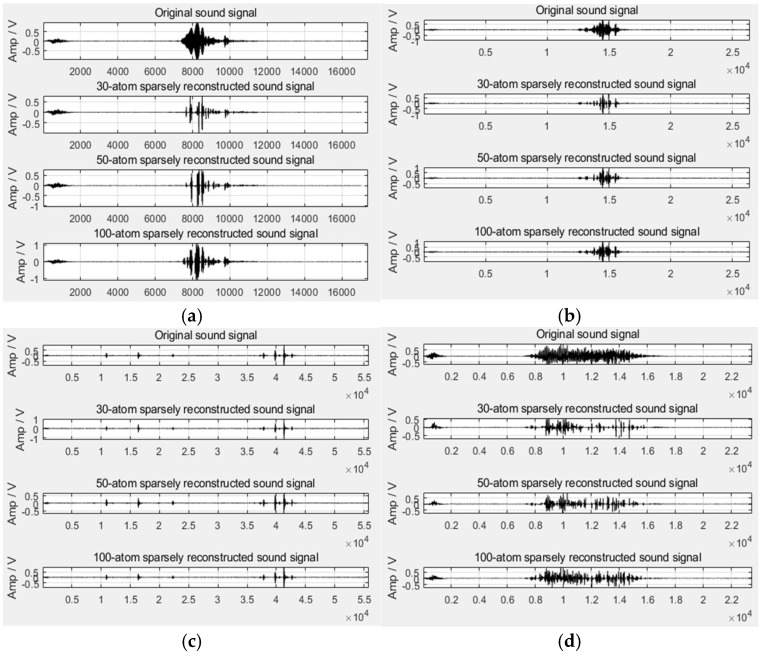
Comparison of sparse reconstruction using different atom numbers of four sound types. (**a**) Crow; (**b**) Cough; (**c**) Purr; (**d**) Flapping wings.

**Figure 7 sensors-22-07935-f007:**
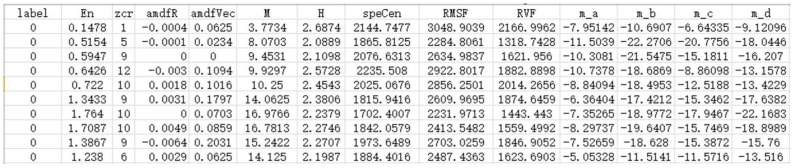
Validation data set to be predicted.

**Table 1 sensors-22-07935-t001:** Specific description of the extracted 60-dimensional classification features.

Number	Feature Name	Symbolic Representation	Number	Feature Name	Symbolic Representation
1	Short-term energy	En	31	9th dimension first-order dynamic feature	dtm_i
2	Short-term average zero-crossing rate	Zcr	32	10th dimension first-order dynamic feature	dtm_j
3	Short-term autocorrelation function	amdfR	33	11th dimension first-order dynamic feature	dtm_k
4	Short-term average amplitude difference	amdfVec	34	12th dimension first-order dynamic feature	dtm_l
5	Short-term average amplitude	M	35	13th dimension first-order dynamic feature	dtm_m
6	Spectral entropy	H	36	1st dimensional second-order dynamic feature	dtmm_a
7	Spectrum centroid	speCen	37	2nd dimension second-order dynamic feature	dtmm_b
8	Root mean square frequency	RMSF	38	3rd dimension second-order dynamic feature	dtmm_c
9	Frequency standard deviation	RVF	39	4th dimension second-order dynamic feature	dtmm_d
10	1st dimension static feature	m_a	40	5th dimension second-order dynamic feature	dtmm_e
11	2nd dimension static feature	m_b	41	6th dimension second-order dynamic feature	dtmm_f
12	3rd dimension static feature	m_c	42	7th dimension second-order dynamic feature	dtmm_g
13	4th dimension static feature	m_d	43	8th dimension second-order dynamic feature	dtmm_h
14	5th dimension static feature	m_e	44	9th dimension second-order dynamic feature	dtmm_i
15	6th dimension static feature	m_f	45	10th dimension second-order dynamic feature	dtmm_j
16	7th dimension static feature	m_g	46	11th dimension second-order dynamic feature	dtmm_k
17	8th dimension static feature	m_h	47	12th dimension second-order dynamic feature	dtmm_l
18	9th dimension static feature	m_i	48	13th dimension second-order dynamic feature	dtmm_m
19	10th dimension static feature	m_j	49	Stretching factor of 30 atomic matching	scale30
20	11th dimension static feature	m_k	50	Translation factor of 30 atom matching	translation30
21	12th dimension static feature	m_l	51	Frequency factor of 30 atom matching	freq30
22	13th dimension static feature	m_m	52	Phase factor of 30 atom matching	phase30
23	1st dimensional first-order dynamic feature	dtm_a	53	Stretching factor of 50 atomic matching	scale50
24	2nd dimension first-order dynamic feature	dtm_b	54	Translation factor of 50 atom matching	translation50
25	3rd dimension first-order dynamic feature	dtm_c	55	Frequency factor of 50 atom matching	freq50
26	4th dimension first-order dynamic feature	dtm_d	56	Phase factor of 50 atom matching	phase50
27	5th dimension first-order dynamic feature	dtm_e	57	Stretching factor of 100 atomic matching	scale100
28	6th dimension first-order dynamic feature	dtm_f	58	Translation factor of 100 atom matching	translation100
29	7th dimension first-order dynamic feature	dtm_g	59	Frequency factor of 100 atom matching	freq100
30	8th dimension first-order dynamic feature	dtm_h	60	Phase factor of 100 atom matching	phase100

**Table 2 sensors-22-07935-t002:** Specific description of the preliminary data set.

Label	Total Number of Data	Number of Training Data	Number of Testing Data
Crow	5918	4143	1775
Cough	4486	3140	1346
Purr	8955	6269	2686
Flapping wing	9532	6672	2860
Total number	28,891	20,124	8667

**Table 3 sensors-22-07935-t003:** Classification accuracies achieved by DT classifier on the data set before and after standardization processing.

Number	Before Processing/%	After Processing/%
1	77.91	77.78
2	77.18	78.49
3	78.36	78.04
4	77.49	77.91
5	77.73	78.30
6	76.34	78.02
7	77.20	77.68
8	77.99	79.32
9	77.78	77.75
10	77.73	78.41
Mean value	77.57	78.17

**Table 4 sensors-22-07935-t004:** Specific description of the retained thirty sound features.

Number	Feature Name	Symbolic Representation	Number	Feature Name	Symbolic Representation
1	Short-term energy	En	16	8th dimension static feature	m_h
2	Short-term average zero-crossing rate	Zcr	17	9th dimension static feature	m_i
3	Short-term average amplitude difference	amdfVec	18	10th dimension static feature	m_j
4	Short-term average amplitude	M	19	11th dimension static feature	m_k
5	Spectral entropy	H	20	12th dimension static feature	m_l
6	Spectrum centroid	speCen	21	13th dimension static feature	m_m
7	Root mean square frequency	RMSF	22	1st dimensional first-order dynamic feature	dtm_a
8	Frequency standard deviation	RVF	23	3rd dimension first-order dynamic feature	dtm_c
9	1st dimension static feature	m_a	24	1st dimensional second-order dynamic feature	dtmm_a
10	2nd dimension static feature	m_b	25	2nd dimension second-order dynamic feature	dtmm_b
11	3rd dimension static feature	m_c	26	3rd dimension second-order dynamic feature	dtmm_c
12	4th dimension static feature	m_d	27	6th dimension second-order dynamic feature	dtmm_f
13	5th dimension static feature	m_e	28	Frequency factor of 30 atom matching	freq30
14	6th dimension static feature	m_f	29	Frequency factor of 50 atom matching	freq50
15	7th dimension static feature	m_g	30	Frequency factor of 100 atom matching	freq100

**Table 5 sensors-22-07935-t005:** Classification accuracies achieved by DT classifier on data sets composed of different number of sound features.

Number	Classification Accuracy/%	
	Sixty	Thirty
1	77.78	81.73
2	78.49	82.09
3	78.04	80.89
4	77.91	80.92
5	78.30	81.88
6	78.02	80.16
7	77.68	81.20
8	79.32	80.58
9	77.75	81.78
10	78.41	81.62
Mean value	78.17	81.29

**Table 6 sensors-22-07935-t006:** Classification accuracies achieved by seven classification algorithms.

Number	SVM/%	Decision Tree/%	Random Forest/%	Naive Bayes/%	kNN/%	BP-NN/%	CNN/%
1	86.78	81.73	90.52	73.04	92.91	64.60	67.18
2	87.20	82.09	90.10	74.61	92.93	64.49	67.22
3	88.14	80.89	90.99	72.91	93.35	64.53	67.19
4	87.25	80.92	90.50	73.27	92.59	64.52	67.27
5	87.72	81.88	90.73	73.64	93.67	64.64	67.27
6	87.36	80.16	90.55	73.95	92.36	64.52	67.23
7	86.13	81.20	90.63	73.04	92.75	64.56	67.20
8	86.36	80.58	90.52	74.53	92.83	64.58	67.25
9	87.96	81.78	90.73	73.72	93.30	64.63	67.26
10	88.06	81.62	90.45	74.71	92.67	64.51	67.24
Mean value	87.30	81.29	90.57	73.74	92.94	64.56	67.23

**Table 7 sensors-22-07935-t007:** Other performance evaluation indexes obtained by five classification algorithms.

Algorithm	Label	Precision/%	Recall/%	*F*1 Value/%
SVM	Crow	78.4	86.9	82.5
	Cough	99.5	84.2	91.4
	Purr	90.6	90.2	90.3
	Flapping wing	88.2	84.1	86.3
decision tree	Crow	73.9	75.4	74.6
	Cough	86.6	83.3	84.7
	Purr	86.2	85.0	85.4
	Flapping wing	79.7	80.4	80.2
random forest	Crow	84.2	88.0	86.1
	Cough	98.4	84.9	91.2
	Purr	92.9	93.2	93.0
	Flapping wing	91.4	90.3	90.9
naive bayes	Crow	69.9	45.9	55.4
	Cough	88.7	51.8	65.3
	Purr	79.5	85.9	82.7
	Flapping wing	67.0	83.1	74.1
kNN	Crow	88.2	90.2	89.1
	Cough	98.2	91.4	94.6
	Purr	95.4	93.8	94.6
	Flapping wing	92.4	93.9	93.1

**Table 8 sensors-22-07935-t008:** Optimal parameter combination of kNN.

Parameter Name	Optimal Value
** *n_neighbor* **	4
** *weights* **	distance
** *metric* **	Euclidean

**Table 9 sensors-22-07935-t009:** Classification accuracies achieved by the classification model based on kNN before and after parameter optimization.

	Number of Testing Data	Number of Data with Correct Predictions	Number of Data with Incorrect Predictions	Average Classification Accuracy/%
Ordinary classification model	8667	8055	612	92.94
Optimal classification model	8667	8161	506	94.16

**Table 10 sensors-22-07935-t010:** Confusion matrix of prediction results.

True Results	Prediction Results	
	Positive	Negative
Positive	*TP*	*FN*
Negative	*FP*	*TF*

**Table 11 sensors-22-07935-t011:** Analysis of prediction results of the classification model.

Number	Number of Data	Numbers of Predicted Crows	Numbers of Predicted Coughs	Numbers of Predicted Purrs	Numbers of Predicted Flapping Wings	Predicted Type	True Type
1	43	0	35	0	8	cough	cough
2	41	39	0	1	1	crow	crow
3	105	0	5	97	3	purr	purr
4	40	1	32	0	7	cough	cough
5	130	14	0	27	89	flapping wing	flapping wing
6	35	0	29	0	6	cough	cough
7	76	2	0	65	9	purr	purr
8	36	31	0	4	1	crow	crow

**Table 12 sensors-22-07935-t012:** Recognition results of ten segments of broiler sound signals.

Number	Number of Sound Types	Number of Predicted Correct Sound Types	Number of Predicted Incorrect Sound Types
1	11	11	0
2	15	15	0
3	11	11	0
4	8	7	1
5	10	10	0
6	12	12	0
7	11	11	0
8	12	12	0
9	13	13	0
10	10	10	0
Total number	113	112	1

## Data Availability

Some or all data, models, or code that support the findings of this study are available from the corresponding author upon reasonable request.

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
