# Peer review of "Recognition Method for Broiler Sound Signals Based on Multi-Domain Sound Features and Classification Model"

_sensors, 2022, doi:10.3390/s22207935_

Round 1

Reviewer 1 Report

This paper proposed a non-contact sound-detection-based broiler health judgment methods. The main contribution of this paper are the feature extraction method and the recognition algorithm. The article has the following problems:

1.   The innovation of the article is insufficient. This article is more about the application of existing methods than original content;

2.   The expression of the article is too long, and the language needs further refinementï¼›

3.   The experimental setup is not reasonable and does not fully reflect the innovation of the article

Author Response

Dear reviewer:

I am very grateful to your comments for the manuscript. According with your advice, we amended the relevant part in manuscript, such as the content marked in red and blue font. Specifically, those paragraphs or statements marked in red are new descriptions added by the author, and those paragraphs or statements marked in blue are descriptions modified by the author. Some of your questions were answered below.

  1. The reviewer’s comment: The innovation of the article is insufficient. This article is more about the application of existing methods than original content;

The authors’ answer: Thank you for your valuable suggestions. We are very sorry that the research content of this article does not satisfy you, but we still have the courage to explain it to you. As you can see in the revised manuscript, in “Introduction”, we have described the background of this study, that is, we hope to achieve automatic broiler health monitoring in the breeding greenhouse through information technology. This is a huge practical need. The broiler sound signal recognition method proposed in this paper is the key to realize the automatic broiler health monitoring. Therefore, all our authors believe that this study is of great practical value. As you pointed out, the author did not propose new technologies or new algorithms in this paper, but combined some technologies or methods in different fields to achieve broiler sound signal recognition. However, we believe that it is an innovation to organically combine technologies or algorithms in different fields, realize some functions and solve some practical problems on this basis. In addition, in “Introduction” of this article, the author also summarizes some contributions of this article, which are listed below. It is worth noting that the author puts forward a new definition of recognition accuracy for broiler sound signals according to the actual needs, which is an innovation in the engineering application field. Therefore, all our authors hope that you can review this article again and consider our explanation. Thank you.

We summarize the contributions of this paper:

(1) Multiple sound features are extracted from the four domains of time domain, frequency domain, MFCC and sparse representation to fully describe the differences between sound types in broiler sound signals.

(2) Use random forest to calculate the importance of each sound feature, and retain those sound features that contribute greatly to the classification effects of the classification model to build a high-quality data set.

(3) Train classification models based on different classification algorithms, compare to obtain the best performer, and optimize its inherent parameters to obtain the optimal classification model.

(4) Combine the prediction results of the classification model and majority voting processing, give the recognition process of sound types, and newly propose the definition of the recognition accuracy for broiler sound signals.

  1. The reviewer’s comment: The expression of the article is too long, and the language needs further refinement;

The authors’ answer: Thank you for your valuable suggestions. As you can see in the revised manuscript, the author spent a lot of time and energy to readjust and describe this article. You can clearly see that the length of the article has been greatly reduced, and many irrelevant paragraphs have been deleted. In addition, some statements that are too long have been modified, and they are all marked in blue font. The author hopes you to review the revised manuscript again. Thank you.

  1. The reviewer’s comment: The experimental setup is not reasonable and does not fully reflect the innovation of the article.

The authors’ answer: Thank you for your valuable suggestions. As you can see in the revised manuscript, the author spent a lot of time and energy to readjust and describe this article. You can clearly see that the length of the article has been greatly reduced, the structure of the article has been readjusted, the experimental setups have been re described, and the innovation of the article has been re-condensed. The author hopes you to review the revised manuscript again. Thank you.

A revised manuscript with the correction sections red or blue marked was attached as the supplemental material and for easy check/editing purpose.

Should you have any questions, please contact us without hesitate.

Thank you and all the reviewers for the kind advice.

Sincerely yours,

Zhigang Sun

E-mail: 2191313@s.hlju.edu.cn

Reviewer 2 Report

In this paper, the author proposes a Classification Feature Extraction and Recognition  Algorithm Implementation of Broiler Sound Signals. Referring to the prediction results, the cough rate can be calculated, and then the broiler health in the area can be judged. It is an important supplement to the existing broiler health monitoring research, and has  extremely high practical value and good application prospects. However, there are some problems in the quality of the paper. The specific problems are as follows:

(1) In Abstract, Please rearrange the abstract, which is confusing.

(2) In Introduction, the author should focus on the highlights of the proposed method and what problems have been solved. such as (1)(2)(3).

(3) At present, there are many classification methods, and the author should analyze the new classification methods in Introduction, such support matrix machine, multi-class fuzzy support matrix machine, Deep stacked support matrix machine method, Twin robust matrix machine and other algorithms.

(4) In the paper, many graphs have low quality. Please improve the quality of graphs.

Author Response

Dear reviewer:

I am very grateful to your comments for the manuscript. According with your advice, we amended the relevant part in manuscript, such as the content marked in red and blue font. Specifically, those paragraphs or statements marked in red are new descriptions added by the author, and those paragraphs or statements marked in blue are descriptions modified by the author. Some of your questions were answered below.

  1. The reviewer’s comment: In Abstract, Please rearrange the abstract, which is confusing.

The authors’ answer: Thank you for your valuable suggestions. As you can see in the revised manuscript, the author restated the abstract, which is listed below.

Abstract: In view of the limited number of extracted sound features, the lack of in-depth analysis of applicable sound features, and the lack of in-depth study of the selection basis and optimization process of classification models in the existing broiler sound classification or recognition research, the author proposes a recognition method for broiler sound signals based on multi-domain sound features and classification model. The implementation process is divided into training stage and testing stage. In the training stage, the experimental area is built, and multiple segments of broiler sound signals are collected and filtered. Through sub-frame processing and endpoint detection, the combinations of start frames and end frames of multiple sound types in broiler sound signals are obtained. A total of sixty sound features from four aspects of time domain, frequency domain, Mel-Frequency Cepstral Coefficients (MFCC) and sparse representation are extracted from each frame signal to form multiple feature vectors. These feature vectors are labeled manually to build the data set. The min-max standardization method is used to process the data set, and the random forest is used to calculate the importance of sound features. Then, thirty sound features that contribute more to the classification effect of the classification model are retained. On this basis, the classification models based on seven classification algorithms are trained, the best performed classification model based on k-Nearest Neighbor (kNN) is obtained, and its inherent parameters are optimized. Then, the optimal classification model is obtained. Test results show that the average classification accuracy achieved by the decision-tree-based classifier (abbr. DT classifier) on the data set before and after min-max standardization processing is improved by 0.6%, the average classification accuracy achieved by the DT classifier on the data set before and after feature selection is improved by 3.1%, the average classification accuracy achieved by the kNN-based classification model before and after parameter optimization is improved by 1.2%, and the highest classification accuracy is 94.16%. In the testing stage, for a segment of broiler sound signal collected in the broiler captivity area, the combinations of start frames and end frames of multiple sound types in the broiler sound signal is obtained through signal filtering, sub-frame processing, endpoint detection and other steps. Thirty sound features are extracted from each frame signal to form the data set to be predicted. The optimal classification model is used to predict the labels of each data in the data set to be predicted. By perform majority voting processing on the predicted labels of the data combination corresponding to each sound type, the common labels are obtained, that is, the predicted types are obtained. On this basis, the definition of recognition accuracy for broiler sound signals is proposed. Test results show that the classification accuracy achieved by the optimal classification model on the data set to be predicted is 93.57%, and the recognition accuracy achieved on the multiple segments of broiler sound signals is 99.12%.

  1. The reviewer’s comment: In Introduction, the author should focus on the highlights of the proposed method and what problems have been solved. such as (1)(2)(3).

The authors’ answer: Thank you for your valuable suggestions. As you can see in the revised manuscript, the author added highlights of the proposed methods and the problems to be solved in “Introduction”. On this basis, the innovation points of this paper are condensed into four points, which are listed below.

It can be seen from the above studies that the current research on broiler sound signal classification is mainly concentrated on the extraction of sound features, including the extraction of sound features that can distinguish different sound types in broiler sound signals, and the extraction of sound features that can distinguish the sounds or calls made by broilers themselves from the environmental noises. On this basis, they built the data set and trained the classification models based on machine learning classification algorithms to complete the recognition of different sound types or different sounds. Although these methods have achieved satisfactory results during the early development process, there still existing three problems. The first is the extraction of sound features. In these studies, they chose to extract one or a few commonly used sound features from time domain or frequency domain from sound signals to build data sets, without considering sound features that can be extracted from multi-domains and extracting more sound features. The second is the analysis of sound features. They did not analyze whether the selected sound features are applicable to the current classification problem. In general, the selected general sound features often perform poorly in dealing with personalized problems. The third is the training of classification model. They chose to train the classification model based on a certain machine learning classification algorithm according to experience, but do not give the selection basis. Meanwhile, they did not optimize the inherent parameters of the trained classification model to further improve its classification effects.

On the basis of summarizing the previous research results, combined with the shortcomings of the existing broiler sound signal classification researches, in this paper, the author proposes a recognition method for broiler sound signals based on multi-domain sound features and classification model. Specifically, in view of the limited sound features extracted in the existing broiler sound signal recognition methods, we extract a total of sixty sound features from four domains, including time domain, frequency domain, Mel-Frequency Cepstral Coefficients (MFCC) and sparse representation. These sound features can fully describe the differences between different sound types in broiler sound signals. In view of the lack of in-depth analysis of sound features in the existing broiler sound signal recognition methods, we use random forest to calculate the importance of sound features and retain those sound features that can effectively reflect the differences among sound types to build the high-quality data set. In order to solve the problem of no in-depth comparison and parameter optimization of classification models in the existing broiler sound signal recognition methods, we train the classification models based on different machine learning classification algorithms, respectively, and compare to obtain the best performer, and optimize its inherent parameters to improve its classification effects. In this way, the optimal classification model can accurately and effectively recognize each sound type in a segment of broiler sound signal, and complete the cough recognition at the same time in this process. This method is an important basis for follow-up research on automatic broiler health monitoring.

We summarize the contributions of this paper:

(1) Multiple sound features are extracted from the four domains of time domain, frequency domain, MFCC and sparse representation to fully describe the differences between sound types in broiler sound signals.

(2) Use random forest to calculate the importance of each sound feature, and retain those sound features that contribute greatly to the classification effects of the classification model to build a high-quality data set.

(3) Train classification models based on different classification algorithms, compare to obtain the best performer, and optimize its inherent parameters to obtain the optimal classification model.

(4) Combine the prediction results of the classification model and majority voting processing, give the recognition process of sound types, and newly propose the definition of the recognition accuracy for broiler sound signals.

  1. The reviewer’s comment: At present, there are many classification methods, and the author should analyze the new classification methods in Introduction, such support matrix machine, multi-class fuzzy support matrix machine, Deep stacked support matrix machine method, Twin robust matrix machine and other algorithms.

The authors’ answer: Thank you for your valuable suggestions. As you can see in the revised manuscript, the author adds analysis and description of new classification methods, including support matrix machine, multi-class fuzzy support matrix machine, deep stacked support matrix machine method, Twin robust matrix machine in “Introduction”. They are described as follows:

It is worth mentioning that, in addition to SVM and decision tree mentioned above, combined with some optimization algorithms, new classification methods also show their skills in solving classification problems in various fields, such as support matrix machine, multi-class fuzzy support matrix machine, deep stacked support matrix machine, Twin robust matrix machine and so on. For example, Xu et al. [15] used support matrix machines, double support matrix machines and near-end support matrix machines to solve the classification problem of two-dimensional image data. Experiments show that the three algorithms have achieved stable and efficient classification results. Horng [16] applied multi-class support vector machine classifier to solve the image classification problem of supraspinatus muscle, and achieved better performance than other implementation methods. By establishing the objective function of non-parallel hyperplane and integrating fuzzy attributes, Pan et al. [17] proposed a multi-class fuzzy support matrix machine (MFSMM), which was used to solve the classification problem of two kinds of roller bearing experimental data. Hang et al proposed a new deep stacked support matrix machine (DSSMM) to improve the performance of existing shallow matrix classifiers in EEG classification. Pan et al. [18] proposed a new non-parallel classifier called twin robust matrix machine (TRMM) and applied it to roller bearing fault diagnosis. Experimental results show that the method has excellent fault diagnosis performance, especially in the presence of abnormal samples.

  1. The reviewer’s comment: In the paper, many graphs have low quality. Please improve the quality of graphs.

The authors’ answer: Thank you for your valuable suggestions. As you can see in the revised manuscript, the author has made a substantial revision to the full text, deleting some pictures and tables of low quality. In addition, if the article can be accepted, the author will also actively cooperate with the editor to provide high-quality vector pictures. Thank you again.

A revised manuscript with the correction sections red or blue marked was attached as the supplemental material and for easy check/editing purpose.

Should you have any questions, please contact us without hesitate.

Thank you and all the reviewers for the kind advice.

Sincerely yours,

Zhigang Sun

E-mail: 2191313@s.hlju.edu.cn

Reviewer 3 Report

This paper proposed a new non-contact sound-detection-based broiler health judgment method, and accordingly carried out this research. The authors extracted 60-dimensional classification features from four aspects of time domain, frequency domain, Mel-Frequency Cepstral Coefficients (MFCC), and Sparse Representation, and applied min-max standardization method to improve the regularity between features. The following review comments are recommended, and the authors are invited to explain and modify.

Comment:  “The classification performance of five classification models were comprehensively analyzed”, why did not authors use deep learning models?

Comment: In intro part, authors should remove irrelevant paras like first and second and make it more concrete.

Comment: Novelty is confusing. A highlight is required. The main contributions of the manuscript are not clear. The main contributions of the ‎article must be very clear and would be better if summarize ‎them into 3-4 precise points at the ‎end of the introduction.‎

Comment: “2. Classification feature extraction”, should be methodology part.

Comment: “3. Recognition algorithm implementation”, should be Experiments and Results part.

Comment: Nothing is mentioned about the implementation challenges.

Comment: The following clinical decision support systems using learning algorithms must be included to improve the quality of the paper.

·       10.1109/ACCESS.2021.3131216

Comment: Discuss the stability of the system in terms of complexity.

Comment: Results are not clearly compared with the state-of-art. This point is fundamental in scientific research.

Comment: Could you please check your references carefully (in particular, proceedings: location of the conference, date of the conference, publisher's name and location...)? All references must be complete before the acceptance of a manuscript.

Author Response

Dear reviewer:

I am very grateful to your comments for the manuscript. According with your advice, we amended the relevant part in manuscript, such as the content marked in red and blue font. Specifically, those paragraphs or statements marked in red are new descriptions added by the author, and those paragraphs or statements marked in blue are descriptions modified by the author. Some of your questions were answered below.

  1. The reviewer’s comment: “The classification performance of five classification models were comprehensively analyzed”, why did not authors use deep learning models?

The authors’ answer: Thank you for your valuable suggestions. As you can see in the revised manuscript, the author added instructions on using the deep learning model in Section 2.4. They are listed below:

Classification algorithms commonly used in machine learning includes SVM [36], decision tree [37], random forest [38], naive bayes [39] and kNN [40]. With the help of performance evaluation indexes, the author applied the above classification algorithms to train classification models on the training set, and took the trained classification models to make predictions on the testing set. In order to obtain relatively accurate prediction results, each classification model make ten times tests to reduce the impact of random errors. It is worth noting that in the research process, the author also drew multiple time-frequency maps representing four sound types to build the picture data set, and on this basis, trained general neural network models, including Back-Propagation neural network (BP-NN) [41] and Convolutional neural network (CNN) [42]. Unfortunately, they did not achieve satisfactory prediction results, and spent more time to obtain these prediction results.

Table 6 lists the classification accuracies achieved by the above seven classification algorithms, and all tests are conducted on the same computer. From the average classification accuracy, we can see that the average classification accuracy obtained by kNN, random forest, SVM, decision tree, naive bayes, CNN and BP-NN decreases successively, which are 92.94%, 90.57%, 88.30%, 81.29%, 73.74%, 67.23% and 64.56%, respectively. Among them, the average classification accuracy obtained by kNN is the highest. The average classification accuracy of random forest is also higher, but there is a certain gap compared with kNN. The classical SVM is mainly used for binary classification, thus the obtained average classification accuracy is slightly lower when it is applied to multi-classification problem. The average classification accuracy obtained by naive bayes is the worst, because naive bayes is a classification method based on the independent assumption of feature conditions. However, in the classification and recognition of broiler sound signals in this research, all the sound features are not completely independent, thus the obtained classification accuracy is poor. The average classification accuracy achieved by the two neural network models is significantly lower than that of the other five traditional machine learning classification algorithms, which shows that the neural network model is not suitable for the broiler sound signal recognition.

Table 6. Classification accuracies achieved by seven classification algorithms.

Number

SVM / %

decision tree / %

random forest / %

naive bayes / %

kNN/ %

BP-NN/%

CNN/%

1

86.78

81.73

90.52

73.04

92.91

64.60

67.18

2

87.20

82.09

90.10

74.61

92.93

64.49

67.22

3

88.14

80.89

90.99

72.91

93.35

64.53

67.19

4

87.25

80.92

90.50

73.27

92.59

64.52

67.27

5

87.72

81.88

90.73

73.64

93.67

64.64

67.27

6

87.36

80.16

90.55

73.95

92.36

64.52

67.23

7

86.13

81.20

90.63

73.04

92.75

64.56

67.20

8

86.36

80.58

90.52

74.53

92.83

64.58

67.25

9

87.96

81.78

90.73

73.72

93.30

64.63

67.26

10

88.06

81.62

90.45

74.71

92.67

64.51

67.24

Mean value

87.30

81.29

90.57

73.74

92.94

64.56

67.23

  1. The reviewer’s comment: In intro part, authors should remove irrelevant paras like first and second and make it more concrete.

The authors’ answer: Thank you for your valuable suggestions. As you can see in the revised manuscript, the author spent a lot of time and energy to readjust and describe this article. You can clearly see that the length of the article has been greatly reduced, the structure of the article has been readjusted, the experimental setups have been re described, and the innovation of the article has been re-condensed. Among them, the first and second paragraphs in “Introduction” you mentioned have been deleted. The author hopes you to review the revised manuscript again. Thank you.

  1. The reviewer’s comment: Novelty is confusing. A highlight is required. The main contributions of the manuscript are not clear. The main contributions of the ‎article must be very clear and would be better if summarize ‎them into 3-4 precise points at the ‎end of the introduction.‎

The authors’ answer: Thank you for your valuable suggestions. As you can see in the revised manuscript, the author added highlights of the proposed methods and the problems to be solved in “Introduction”. On this basis, the innovation points of this paper are condensed into four points, which are listed below.

  1. The reviewer’s comment: “2. Classification feature extraction”, should be methodology part. “3. Recognition algorithm implementation”, should be Experiments and Results part.

The authors’ answer: Thank you for your valuable suggestions. As you can see in the revised manuscript, the author spent a lot of time and energy to readjust and describe this article. You can clearly see that the length of the article has been greatly reduced, the structure of the article has been readjusted, the experimental setups have been re described, and the innovation of the article has been re-condensed. Among them, the two parts you mentioned have been readjusted in the revised manuscript. The author hopes you to review the revised manuscript again. Thank you.

  1. The reviewer’s comment: Nothing is mentioned about the implementation challenges.

The authors’ answer: Thank you for your valuable suggestions. As you can see in the revised manuscript, the author described many implementation challenges during the development of this study, including the broiler sound signal collection, manually labeling the data, and the wrong recognition of sound types. It is worth noting that some implementation challenges have been solved and explained in the article, and some implementation challenges have been described in “Discussion” of the article and listed as future research plans. Some of them are listed below.

It should be noted that when the author directly arranged the audio collection system in the broiler captivity area in the breeding greenhouse, because the broilers are afraid of strangers and electronic equipment, the broilers close to the audio collection system would scream or flap their wings. These abnormal and high-frequency sounds would quickly affect other broilers around, causing the collected signals not truly reflect the growth conditions of the broilers in the captivity area. Under normal growth conditions, the broilers are in a quiet state, and individual broilers occasionally crow, cough, etc., and are not contagious. However, most of the sound signals collected in this abnormal growth conditions are overlapping broiler sound signals, which is not conducive to the early development of this research. Therefore, the author built an experimental area far away from the broiler captivity area, in order to collect continuous single broiler sound signals without overlapping. This is friendly and helpful for the early development of this study.

It should be noted that the ten times predictions performed by the classification model in “Verification and Analysis” showed that, a purr was misjudged as a cough in the fourth segment of sound signal, because the two sounded similar. But the difference between the two is that the former is a persistent and continuous process, which means that a purr will eventually get more frame signals, while the latter is a relatively short process, which means fewer frame signals. In the following research, the author used the audio collection system to obtain multiple segments of broiler sound signals again, and also constructed multiple data sets to be predicted through signal filtering, sub-frame processing, endpoint detection and feature extraction. The author used the optimal classification model based on parameter-optimized kNN to make predictions on these data sets, respectively, and combined the majority voting processing to obtain multiple recognition results. The recognition results also show that the classification model has reliable recognition effects, and there is no case that the cough is mistaken for the other three sound types. This shows that the classification model has a high sensitivity to the recognition of coughs, which is conducive to the follow-up research on automatic broiler health monitoring. That is, using the currently trained classification model will not miss or incorrectly recognize the coughs in the broiler sound signal, which is extremely important for the stable judgment of broiler health.

  1. The reviewer’s comment: The following clinical decision support systems using learning algorithms must be included to improve the quality of the paper.

10.1109/ACCESS.2021.3131216

The authors’ answer: Thank you for your valuable suggestions. As you can see in the revised manuscript, the author added a description of the clinical decision support system in “Discussion” of the article. They are listed below.

It is worth noting that Qadri et al. [49,50] advocate the clinical decision support system using learning algorithm to further improve or reflect the application value of the proposed method. Therefore, shortly after the completion of this study, the author began to design the audio collection terminal placed in the broiler captivity area and visualization software. At present, the audio collection terminal and visualization software have been designed and preliminarily debugged. On this basis, the combination of audio collection terminal, broiler sound signal recognition method and visualization software can build a clinical decision support system for automatic broiler health monitoring mentioned in “Introduction”. That is, the audio collection terminal obtains the broiler sound signals in the broiler captivity area and transmits it to the remote computer in a timely manner through wireless communication. Broiler sound signal recognition method completes the task of recognizing different sound types in broiler sound signals, and saves the recognition results locally. The visualization software reads the recognition results for display. This part is also the author’s future research content.

  1. The reviewer’s comment: Discuss the stability of the system in terms of complexity.

The authors’ answer: Thank you for your valuable suggestions. As you can see in the revised manuscript, the author verified the broiler sound recognition method proposed in this paper from two aspects in “Verification and analysis”. On the one hand, it is the classification effects of the classification model, and on the other hand, it is the recognition effects of the classification model. The whole verification process is to discuss the feasibility and stability of the proposed method from the perspective of complexity. It is worth noting that when verifying the recognition effects of the classification model, we directly collected the broiler sound signals to be predicted in the broiler captivity area. These signals can more feedback the complex and real broiler growth environment. Therefore, all our authors hope you to review the revised manuscript again and understand our instructions. Thank you.

  1. The reviewer’s comment: Results are not clearly compared with the state-of-art. This point is fundamental in scientific research.

The authors’ answer: Thank you for your valuable suggestions. As you can see in the revised manuscript, the author has proposed a new definition of recognition accuracy for broiler sound signals, which was not found in public-published articles. Therefore, it is difficult for the author to directly compare the recognition accuracy obtained with that obtained by other scholars, because their calculation principles are different. As you know, the recognition accuracy obtained by many scholars is actually the classification accuracy in machine learning, that is, the prediction accuracy of a classifier or classification model for the labels of each piece of data in a data set. This is at the data set level, not in line with the actual engineering application needs. If the author compares the classification accuracy obtained by the most advanced technology with the classification accuracy obtained by the optimal classification model in this paper, the essence is to compare the classification effects of the classifier. In fact, this comparison is meaningless. Moreover, the classification effects of classifiers can be continuously improved. The comparison between different classifiers also has low reference value. Therefore, from the perspective of recognition accuracy, the author did not find the most advanced technology to compare. Of course, the retrieval scope of the author may be limited. If you have the most advanced technology to provide, we will be grateful and supplement this part in the next revision. All our authors sincerely hope you to review the revised manuscript and accept our instructions. Thank you again for your valuable suggestions.

  1. The reviewer’s comment: Could you please check your references carefully (in particular, proceedings: location of the conference, date of the conference, publisher's name and location...)? All references must be complete before the acceptance of a manuscript.

The authors’ answer: Thank you for your valuable suggestions. As you can see in the revised manuscript, the author has added new references and revised the original references to ensure that all information is complete. The basic information required for the conference paper you mentioned has been provided in the revised manuscript.

A revised manuscript with the correction sections red or blue marked was attached as the supplemental material and for easy check/editing purpose.

Should you have any questions, please contact us without hesitate.

Thank you and all the reviewers for the kind advice.

Sincerely yours,

Zhigang Sun

E-mail: 2191313@s.hlju.edu.cn

Round 2

Reviewer 3 Report

Authors answered satisfactory.